# UAV’s Status Is Worth Considering: A Fusion Representations Matching Method for Geo-Localization

**DOI:** 10.3390/s23020720

**Published:** 2023-01-08

**Authors:** Runzhe Zhu, Mingze Yang, Ling Yin, Fei Wu, Yuncheng Yang

**Affiliations:** School of Electronic and Electrical Engineering, Shanghai University of Engineering Science, Shanghai 201602, China

**Keywords:** cross-view image matching, geo-localization, UAV image localization, multimodal, transformer, bilinear pooling, deep learning

## Abstract

Visual geo-localization plays a crucial role in positioning and navigation for unmanned aerial vehicles, whose goal is to match the same geographic target from different views. This is a challenging task due to the drastic variations in different viewpoints and appearances. Previous methods have been focused on mining features inside the images. However, they underestimated the influence of external elements and the interaction of various representations. Inspired by multimodal and bilinear pooling, we proposed a pioneering feature fusion network (MBF) to address these inherent differences between drone and satellite views. We observe that UAV’s status, such as flight height, leads to changes in the size of image field of view. In addition, local parts of the target scene act a role of importance in extracting discriminative features. Therefore, we present two approaches to exploit those priors. The first module is to add status information to network by transforming them into word embeddings. Note that they concatenate with image embeddings in Transformer block to learn status-aware features. Then, global and local part feature maps from the same viewpoint are correlated and reinforced by hierarchical bilinear pooling (HBP) to improve the robustness of feature representation. By the above approaches, we achieve more discriminative deep representations facilitating the geo-localization more effectively. Our experiments on existing benchmark datasets show significant performance boosting, reaching the new state-of-the-art result. Remarkably, the recall@1 accuracy achieves 89.05% in drone localization task and 93.15% in drone navigation task in University-1652, and shows strong robustness at different flight heights in the SUES-200 dataset.

## 1. Introduction

Unmanned aerial vehicles (UAVs) have been developed rapidly over the past few decades, and the flexibility and stability of UAVs are continuously improved. Currently, they are widely used in national defense, agriculture, mapping, and other fields [1,2,3,4]. Hassan et al. [5] detect and count rice plants by using an unmanned aerial vehicle. Oh et al. [6] proposed an autonomous UAV smart search system method to search for persons in disaster sites. However, positioning UAVs in the GNSS-denied environment is still a significant challenge. Visual geo-localization technology is the key to solving this problem. Geo-localization aims to match an image from one perspective to the most similar image from another perspective that represents the same geographic target. There are two main tasks for drones: 1. Drone Localization 2. Drone Navigation. As the name shows, drone localization intends to predict the location of the target place via drone-view images. On the other hand, given a satellite-view image as a query image, drone navigation is to drive the drone to the area of the target in the query. Cross-view image matching is still a challenging task due to the viewpoints between drone image and satellite image are different. Moreover, their image styles can also be far apart because their from various source. As a result, finding the corresponding image in another view can be really challenging. In recent years, deep learning-based methods have been applied to geo-localization, showing competitive results. Existing deep neural network regards the geo-localization task as an image retrieval task, and the aim of the network is to make the corresponding image pair closer to each other in high dimension space. Most conventional approaches focused on applying spatial attention mechanisms to extract geographic features. Those features describe the main target or contextual information around the target, but they neglect the extra information from the drone itself. In fact, the drone’s flight status has high relevance to the image capture. In addition, average pooling is used to extract global features after the backbone network, ignoring the interaction between inter-layer local features and global features. These shortcomings make previous methods not robust enough when facing diverse scenes and environmental conditions. Our method mainly establishes two modules to address those problems. Firstly, we apply a multimodel to consider the status of the drone, such as flight heights, and camera pose. The potential status’ information is collected from an existing geo-localization dataset and transformed into word embeddings by BERT. To help the model learn other dimension data, we adopt progressive fusion to link image embeddings and word embeddings. Secondly, we develop an HBP-based feature aggregation module to extract local-aware features. Precisely, our HBP module fuse local and global features from cross-layer to reinforce each other to distinguish similarly. Consequently, our extensive experimental results demonstrate that our method achieves superior geo-localization performance to the state-of-the-art. Remarkably, the recall@1 accuracy achieves 89.05% in the drone localization task and 93.15% in the drone navigation task in University-1652. It shows strong robustness at different flight heights in the SUES-200 dataset, with only a 6.5% decrease in drone localization task and 7.5% decrease in drone navigation task in SUES-200 dataset. The code is available at https://github.com/Reza-Zhu/MBF (accessed on 24 November 2022).

### 1.1. Related Work of Geo-Localization Methods

In the early stage, most geo-localization studies were based on hand-craft features to find the most relevant image [7,8,9,10,11]. However, these methods are less robust and susceptible to unfavorable factors such as lighting and occlusion. With the development of deep learning, neural networks have replaced hand-craft methods. Lin et al. [12] built a dataset of 78K aligned cross-view image pairs and proposed an approach named Where-CNN. Tian et al. [13] mainly collected lots of locations in the city, constructed image pairs using the “bird” view and the street view, and then presented a Siamese network to find the correct target for each query image. CVUSA [14] is a standard cross-view dataset consisting of panoramic street view and satellite view image pairs. Many previous works have been conducted based on this dataset. CVACT [15] is a larger panoramic dataset than CVUSA, with improved satellite image resolution and several test sets. Liu et al. [15] also designed novel network endowed deep neural networks with commonsense orientation. Some other works focused on the matching between drone-view and satellite-view. Zheng et al. [16] innovatively collected a new dataset, University-1652, which contains street-view, drone-view, and satellite-view images. Because of University-1652, cross-view geo-localization achieved remarkable progress in the past two years. Ding el at. [17] proposed LCM, which utilized ResNet [18] as the backbone network and trained the image retrieval problem as a classification problem. LCM also added a data augmentation method to balance the number of drone-view and satellite-view images. Wang el at. [19] designed LPN after considering the contextual information of neighboring regions, which applied the square-ring partition strategy to divide feature maps. Zhuang et al. [20] proposed MSBA based on a local pattern network designed to make different branches learn from each other. Tian et al. [21] adopted conditional generative adversarial nets to synthesize the drone-view image with a vertical style. Transformer architectures have obtained spectacular success in recent years, Dai et al. [22] used ViT [23] as the model’s backbone to extract images’ context information and divided regions based on heat distribution of feature map (FSRA), and then FSRA aligned multiple specific areas into a set of feature representations. In geo-localization, they can be subdivided into two groups including Street↔Aerial, and Drone↔Satellite. Table 1 gives an overview of existing approaches.

### 1.2. Related Work of Multimodal

With the development of multimodal, progress has been made in the past year in deep learning [26,27,28]. Tan et al. [29] proposed the LXMERT framework to solve the VQA task, which understood the connections between visual concepts and linguistic semantics. For large-scale datasets, Radford et al. [30] collected nearly 400 million image text pairs and designed two encoders to extract their features (CLIP). The result showed CLIP had strong robustness in zero-shot classification and downstream vision tasks. The fusion of features can be divided into the following categories in practical works: (1) early fusion, which means multi-modality data would be fused before inputting model [31]. (2) late fusion or feature-level fusion. It gathers the multi-modality features extracted by the corresponding branch to obtain the final feature representation [32]. (3) progressive fusion. It fuses the multi-modality features progressively instead of at once, which can improve the positive effects of complementary information and better aggregate details from each modality [33].

### 1.3. Related Work of Pooling

The aggregation of representation is another significant spot in computer vision. Previously, some classical approaches have been used, e.g., max pooling [34], average pooling [35], and hybrid pooling [36]. Radenovic et al. [37] proposed a novel pooling layer based on generalized-mean (GeM) that has learning parameters to fuse features. In order to obtain an informative image descriptor, Lin et al. [38] designed bilinear models that can model pairwise features, which is especially useful for fine-grained categorization. Gao et al. [39] developed compact bilinear pooling (CBP) to extract representations through a kernelized analysis method. Fukui et al. [40] relied on CBP [39], and proposed multimodal compact bilinear pooling (MCB) to combine multimodal features and applied MCB on the visual question answering. Yu et al. [41] presented a hierarchical bilinear pooling (HBP) framework to integrate multiple cross-layer bilinear features to enhance their representations capability.

### 1.4. Contributions

To fill the existing gaps, we mainly made the following contributions:

1. Different from former geo-localization works, which are based on a single architecture backbone. We divide the backbone into two parts; the first part adopts ResNetV2 [42] to learn fine-grained features, and the second part uses Vision Transformer (ViT) [23] to extract global features from images.

2. We introduce a pioneering multimodal method for geo-localization. To fuse extra vehicle information, e.g., flying height, camera angle, etc., we treat them as multi-modality data and use a sentence to describe that information. Then, that semantic information is converted into a word embedding and learned by the network with image embeddings.

3. A feature fusion mechanism based on HBP is proposed to integrate local parts and global features from the same viewpoint. Specifically, the fusion module fuses cross-layer representations in a mutually reinforced way, improving the robustness of the final feature.

The remainder of the paper is organized as follows: Section 2 describes the datasets applied in the experiments. Section 3 introduces the implementation of multimodel and bilinear pooling and presents the structure of the MBF. Section 4 displays the experiments and results. The discussion and conclusion are given in Section 5 and Section 6.

## 2. Datasets

### 2.1. University-1652

One dataset used in this study is University-1652, collected by Zheng et al. [16]. University-1652 contains 1652 locations from 72 universities. Each site has several street-view images, 54 drone-view images, and one satellite-view image. The street images were collected from Google Maps and the image search engine. For drone-view images, because it was unaffordable to collect a large number of real drone-view images, the authors leveraged the 3D models provided by Google Earth to simulate the actual drone flying (height of view from 256 m to 121.5 m). Satellite-view images with GNSS information were from Google Maps.

In our study, the matching between drone-view and satellite-view is selected as the main tasks. Specifically, we denote the above two tasks as: 1. Drone→Satellite. 2. Satellite→Drone. The dataset is split into the training set and the testing set. There are 701 target locations in the dataset for learning or query and 951 for reference. The drone flight path is shown in Figure 1a. The details of University-1652 are shown in Table 2.

### 2.2. SUES-200

Another dataset implemented in experiments is SUES-200, released by Zhu et al. [43]. SUES-200 is also a cross-view matching dataset with only two views, drone-view and satellite-view. It pays attention to the flying height of the drone and selects a boarder range of scene types. There are four heights in SUES-200, 150 m, 200 m, 250 m, and 300 m. In addition, the camera angles at the four heights are 45°, 50°, 60°, and 70°. Concretely, SUES-200 collected 50 images for drone-view and one image for satellite-view at each location. The drone-view images were captured by the drone in the real world, and the flight trajectory is one circle around the target scene with a fixed height. The satellite images were collected from AutoNavi and Bing Maps in corresponding target locations.

We also denote two tasks as 1. Drone→Satellite. 2. Satellite→Drone. There are 120 locations in SUES-200 for training and query and 80 locations for reference. The drone flight path is shown in Figure 1b. The statistics of the dataset are shown in Table 3.

### 2.3. Evaluation Indicators

University-1652 and SUES-200 both support Recall@K(R@K) and average precision(AP). Therefore, R@K and AP are mainly used to evaluate the performance of our model.

R@K is very sensitive to the position of the first true-matched image appearing in the ranking of the matching result. Therefore, it is suitable for a test dataset that contains only one true-matched image in the candidate gallery. The equation for R@K is presented as follows:(1)$$\mathrm{Recall}@K=\left\{\begin{array}{c} \;1,\;if\;order_{true}<K+1\\ \;0,\;otherwise \end{array}\right.$$
curve, which considers the position of all true-matched images in the evaluation. The equations for AP are shown as follows: (2)$$\mathrm{AP}=\frac{1}{m}\sum_{h=1}^{m}\frac{p_{h-1}+p_{h}}{2},\;\mathrm{where}\;p_{0}=1,\;\;p_{h}=\frac{T_{h}+1}{T_{h}+F_{h}}$$
where *m* is the number of correct matching images, $T_{h}$ and $F_{h}$ are the numbers of correct matching images and incorrect matching images before the (i+1) correct matching image.

## 3. Method

### 3.1. Backbone

Because we concentrate on drone-view and satellite-view matching, the proposed model, MBF, contains two main branches, which is similar to the Siamese network [44]. Most mainstream methods applied ResNet-50 [18] as their backbone network and designed an effective attention mechanism to extend the model’s contextual information extraction capacity. The newly elaborated backbone differs from previous practice in that we employ Hybrid ViT [23] architecture as the model’s feature extractor. Specifically, MBF’s backbone has two shared-weight branches, and each branch’s structure consists of ResNetV2 [42] and ViT [23]. ResNetV2 demonstrated the significance of identity mapping, He et al. [42] proposed a new residual block that moved the activation function and batch normalization layer before the convolution layer. In MBF’s Backbone, ResNetV2 is designed to have three bottlenecks. Each bottleneck owns 3, 4, and 9 residual blocks, respectively. Following, we employ ViT to process the feature map, the patch layer partitions the map into patch embeddings, and the Transformer layer extracts the contextual semantic relationship between each embedding.

In visual geo-localization, the ResNet-based network can capture the target’s fine-grained feature from a small-scale dataset. Researchers usually add attention modules after the backbone to enforce the model’s global feature extraction ability. A ViT-based network delivers a more comprehensive solution due to patch embeddings and self-attention mechanism. However, training a ViT-based model highly relies on large-scale datasets, for example, ImageNet. Therefore, we select a Hybrid ViT as MBF’s backbone. It shows powerful representation extraction ability and expansibility in cross-view matching.

The overview structure of MBF is shown in Figure 2, apart from the backbone structure. The Multimodal block will be detailed in Section 3.2. The HBP block will be described in Section 3.3. The Feature Partition Strategy(FPS) and Loss function will be detailed in Section 3.4.

### 3.2. Multimodal

Multimodal deep learning has achieved great success in recent years, e.g., Image Text Retrieval, and Visual Grounding. Moreover, the proposal of Transformer [45] gives the community a novel direction due to Transformer’s expansibility. In geo-localization, some methods [15,21] also tried to use extra information beyond the image, e.g., orientation, and spatial position, to boost the matching performance. The question is how to introduce other details such as flying height and camera angle though an end-to-end manner?

Intuitively, we can concatenate that information with an image feature map or input it directly to the backbone. However, we have to redesign the network according to different data forms. In MBF, we present a more elegant way. In short, we collect extra data from existing datasets and merge them into the model. Taking University-1652, for example, it details the drone’s flying height in the paper [16]. Hence, we describe that information in a sentence. Referring to SUES-200 presents not only the height but also the camera’s shot angle, but a sentence can still easily cover that information.

To align the feature’s dimension with the image’s patch embedding, we utilize a pre-trained NLP model BERT [46] to transform the sentence to the word embedding. Furthermore, the timing of fusing the word embedding to the backbone should be considered. As shown in Figure 3, we adopt progressive fusion to link the text model data. The purpose of this module is to help the network learn external information in an implicit way. Fusion experiments will be detailed in Section 4.3.

### 3.3. Bilinear Pooling

An adequate representation fusion method is needed because of the importance of the feature’s robustness and discrimination. Inspired by research findings from Personal ReID [47], the bilinear pooling is applied to fuse the feature maps. MBF’s backbone has two stages, ResNetV2 focuses on the target’s local feature, and ViT focuses on the global feature. We designed a hierarchical feature fusion block based on HBP to enhance the final feature’s capability.

As shown in Figure 2, we add a Transformer block at the ending of ViT. There are three feature maps from different training phrases. We perform a pairwise fusion of those features between global and local features. There are two steps in HBP’s pipeline, bilinear transformation, and spatial global pooling. In the first step, we denote $f_{j}$ as extracted feature of input image $x_{j}$ and $F_{conv}$ as CNN block, that contains the Convolution layer, Batch Normalization layer, and Relu layer, as in Figure 4. The formula is calculated as follows:(3)$$f_{j}=F_{conv}\left(f_{j}\right)$$

In the second step, where $f_{j}^{1}$ is the global feature, $f_{j}^{2}$ is the local feature, × stands as the outer product of two vectors:(4)$$f_{x,y}^{p}=f_{j}^{1}\times f_{j}^{2}$$
(5)$$f_{x}^{p}=\sum_{y=1}^{n}f_{x,y}^{p}$$

Then, the fusion feature $f_{x}^{p}$ is normalized to generate the final feature vector $\tilde{f}$, and $\epsilon =1\times 10^{-12}$:(6)$$f_{x}^{p}=\sqrt{f_{x}^{p}}$$
(7)$$\tilde{f}=\frac{f_{x}^{p}}{\max(∥f_{x}^{p}∥,\epsilon )}$$

### 3.4. Improvement and Loss Function

To maximize MBF’s potential, we apply FPS to explicitly take advantage of contextual information. LPN [19] has shown its breakthrough results in the field of cross-view geo-localization. However, unlike LPN’s part number, we deploy n=2 because MBF’s backbone has paid attention to contextual information.

Each part of the features is followed by an MLP block, and softmax processing is performed on the output of that block to normalize the value range from 0 to 1. We adopt cross-entropy as the loss function, which is typical in multi-classification tasks. The cross-entropy is mainly used to determine how close the actual output is to the expected output. $z_{j}^{i}\left(y\right)$ is the logarithm of ground-truth *y*, and $\hat{p}\left(y|x_{j}^{i}\right)$ is the probability of the predicted outcome of the model equal to ground-truth *y*. The mathematical formula is shown:(8)$$\hat{p}\left(y|x_{j}^{i}\right)=\frac{exp(z_{j}^{i}\left(y\right))}{\sum_{c=1}^{C}exp\left(z_{j}^{i}\left(c\right)\right)}$$
(9)$$Loss_{ce}=\sum_{i,j}-log\left(\hat{p}\left(y|x_{j}^{i}\right)\right)$$

When testing the model’s accuracy, cosine distance is used to judge the similarity between features. Triplet loss is widely used in deep metric learning as a supervisor to narrow the distance between the same targets from different backgrounds. In the cross-view matching task, drone-view and satellite-view images describe the same geographical location from different views. In order to differentiate them between similar scenes, Triplet loss is applied in MBF to learn discriminative features. Where *d* denotes the distance between two feature, *a* stands anchor sample, *p* and *n* represents positive sample and negative sample, respectively, *M* is the value of margin. The mathematical formula is shown:(10)$$d\left(x,y\right)=\sqrt{\sum_{i=1}^{n}{\left(x_{i}-y_{i}\right)}^{2}}$$
(11)$$Loss_{triplet}=max\left(d\left(a,p\right)-d\left(a,n\right)+M,0\right)$$

In the two-branch backbone network, both outputs of the model need to be compared with the label. We let the loss of drone-view be $L_{d}$ and the loss of satellite-view be $L_{s}$. These two loss values are added to obtain $L_{total}$. We optimize the whole network through $L_{total}$. The equation expression is shown as follows: (12)$$Loss_{total}=L_{s}+L_{d},$$

## 4. Experiments

### 4.1. Implement Details

We employ a Hybrid ViT (ResNetV2-50 and ViT-Base) pre-trained on ImageNet as MBF’s backbone initialization weight. We resize each input image to a fixed size 384×384 pixels and perform image augmentation, e.g., random crop, random perspective, random affine, and random horizontal flip. In training, SGD is chosen as the optimizer with a momentum of 0.9 and weight decay of $5\times 10^{-4}$ with a mini-batch of 16. The initial learning rate is 0.01 for the backbone layer and 0.1 for the classification layer, it decays by 0.25 every 25 epochs, and the drop out rate is 0.35. In the inference stage, the cosine distance is utilized to measure the similarity between the query image and candidate images in the gallery. Our model is built basing Pytorch, and all experiments are conducted on an NVIDIA RTX Titan GPU.

### 4.2. Comparison with the State-of-the-Art

As shown in Table 4, we compare the proposed method with other methods on University-1652. MBF has achieved 89.05% Recall@1 accuracy, 90.61% AP on Drone→Satellite, 93.15% Recall@1 accuracy and 88.17% AP on Satellite→Drone with 384 × 384 image input size. The performance of MBF has greatly surpassed that of the existing state-of-the-art model and outperforms the best method, MSBA with about 3% overall improvement.

For additional evaluation, in Table 5, we also test MBF on SUES-200. In Drone→Satellite, compared with LPN [19], the Recall@1 accuracy increases from 61.58%, 70.85%, 80.38%, and 81.47% to 85.62%, 87.43%, 90.65%, and 92.12% in four heights. In Satellite→Drone, The Recall@1 accuracy raises from 83.75%, 88.75%, 92.50 %, and 92.50% to 88.75%, 91.25%, 93.75%, and 96.25% in four heights with 384 × 384 image input size. The result to emerge from the data is that MBF has excellent and stable performance at different flight heights.

### 4.3. Ablation Study of Methods

#### 4.3.1. Effect of the Fusion Timing

In multimodal, the timing of fusion features from different modals is vital [50]. We divide the opportunity of fusion into early, progressive, and late fusion. However, the early fusion has to discard enormous amounts of information to align dimensions. Figure 5 shows the network architectures. Hence, we test progressive fusion and late fusion in our experiments. As shown in Table 6, late fusion obtains higher Recall@1 accuracy and AP.

#### 4.3.2. Effect of the HBP Design

To find the best representation fusion form, we design it in three different ways. Figure 6 shows those architectures. We carry out the following comparative experiments to study the influence of different HBP designs. The matching accuracy is set to be our evaluation metric. According to Table 7, it is clear that when Triplet HBP fusion form is settled as our method, the model has the highest matching accuracy.

#### 4.3.3. Effect of the Part Feature Numbers

The number of part features is an essential indicator in PFS. When n=1, the network will use the full feature map. In the experiments, we change the number of parts for the feature map, i.e., 1, 2, 3, and 4. The matching results can be seen in Table 8. We note that the parts cannot improve the quality of the final feature map as we intuitively think. When n=3,4, the performance gains slightly degrades. Therefore, we set n=2 for our model.

#### 4.3.4. Effect of the Proposed Methods

As shown in Table 9, methods ablation experiments for methods are performed. In those methods, Multimodal and HBP are two main tricks to improve the performance, contributing nearly 3% for both tasks. Additonally, the FPS still plays a significant role in our approach, which helps the model enhance 2% matching accuracy. Compared with the baseline model, the methods proposed in this paper provide +6% Recall@1 and +5% AP on the task Drone→Satellite, and +3% Recall@1 and +4% AP on the task Satellite→Drone. The experimental results demonstrate the efficiency of Multimodal and HBP; MBF can grasp more information from various data sources, and a well-designed pooling block also extracts powerful features.

### 4.4. Shifted Query

To verify the performance of MBF in the natural environment, we use flip padding transform on query images for testing. Flip padding flips *n* pixels on the left side of the images and crops *n* pixels on the right side of the image(See Figure 7). This process explores whether MBF could cope with the offset of the target location in the true-matched image pair. Table 10 shows the experimental results. Additionally, 0 indicates the original image (384 × 384), and 10, 20, 30, and 40 are the shifted pixels *n*, respectively. The results show that MBF can tolerate the sightly distraction of the query image. However, when the query image is shifted by more pixels, both tasks’ performance decreases more.

### 4.5. Multi Query

It is hard to comprehensively describe the target location with a single drone-view image. However, University-1652 provides images from different viewpoints and angles for each area. This means that for Drone→Satellite, we can use multiple drone-view images as queries simultaneously to explore whether these multi-view queries can improve the performance of matching. We average the features obtained from multiple images in the evaluation and adopt them as a query. The results can be seen in Table 11. We set the number of images to 54, 27, 18, 9, 3, 2, 1, and the experimental outcomes clearly show that the matching accuracy of multiple queries can be greatly improved by about 5% in both Recall@1 and AP. It also shows that multi-angle features are beneficial for drone localization task in future practice.

Furthermore, we also compared our results with other competitive methods, as shown in Table 12. It can be seen that the AP value of our method is 3% higher than other methods in University-1652.

### 4.6. Inference Time

In actual application, inference speed is an important evaluation metric for geo-localization algorithms. Due to the default, the image features in the gallery set are stored locally in advance in the inference phase. Taking Task 1 (Drone→Satellite) for an example, satellite-view images are processed in advance and stored in memory. Therefore, there are two main steps of inference: 1. The process of extracting query (drone-view) image feature. 2. Calculate the distances between query (drone-view) feature and local gallery (satellite-view) features. We test the inference time on an NVIDIA RTX TiTAN GPU. As shown in Table 13, MBF needs to spend 2.56 s to infer one query image and $3.61\times 10^{-4}$ s to calculate distance. Compared with ViT-Basee and LPN, it has to take longer time which is still hard to apply in the large-scale dataset.

### 4.7. Visualization

In this section, we visualize the retrieval result of MBF on test datasets from University-1652. Figure 8 shows the visualization results under rank 5 in two tasks, MBF could correctly retrieve in some very similar scenes.

The heat map, Figure 9, is generated by MBF from University-1652’s testing set. We can easily observe that MBF’s activation areas contain the target and background areas compared with the previous method. MBF shows its strong ability in discriminative feature extracting and global attention.

## 5. Discussion

In our study, we mainly explore two questions, how to introduce external information to the network in a multimodal manner and how to fuse cross-layer representations. Compared with other methods, they have tried to design appropriate approaches to mine valuable information only from images. Our approach proceeds through solving the above two questions to mine data not only from the image but also from the exterior, for example, the vehicle itself. First, MBF adopts Hybrid-ViT as its backbone, combining the merits of CNN and Transformer architecture. Secondly, we use a sentence to describe the status of the drone and satellite. BERT is applied to convert the semantic information to the word embedding to input the model. However, the descriptions of the vehicle are still limited by the dataset. Thirdly, HBP is used to fuse local and global features from different layers, and we note that it deals with the problem of lack of feature robustness well and shows enormous potential value in geo-localization. The experiments of this study demonstrate our proposed method could learn more potential information in the corresponding dataset, and the matching accuracy is also greatly improved. However, we also observe that the matching accuracy of Satellite→Drone is hard to improve due to some very similar scenes being hard to tell apart. This is a direction in which we will conduct further research.

In addition, the multimodal application in geo-localization is still in its early stages and it is needed to design more studies and dig deeper. For example, multimodal datasets are still very lacking. The community also needs cleverly designed networks to incorporate more information. We hope our approach could serve as a modest spur to induce someone to come forward with their valuable contributions.

Future work will focus on collecting a comprehensive multimodal geo-localization dataset for the community. This dataset contains more information about drone status, satellite status, and even weather conditions. Then, we could apply multimodal-based model on it to further improve the performance of model in various environments. On the other hand, the inference time of current model is another big concern. We also will work on reducing the parameters of backbone to faster the inference speed.

## 6. Conclusions

In this paper, we focused on visual geo-localization. It could be applied in GNSS-denied situations or complex electromagnetic environments to help drones compete in navigation and localization tasks. Through a summary of previous studies, we proposed a method that has proved the feasibility of multimodal’s application in the geo-localization field and the reliability of feature fusion based on HBP. MBF absorbs valid data through the multimodal paradigm and improves the feature map’s quality by reinforcing representations.

The experimental result validates our method, MBF has achieved state-of-the-art results on existing datasets, i.e., University-1652, SUES-200. Moreover, the proposed MBF showed the toleration of shifted query images, which is close to practical application. The multiple queries also prove the importance of comprehensive viewpoints when the drone performs localization tasks.

## Figures and Tables

**Figure 1 sensors-23-00720-f001:**
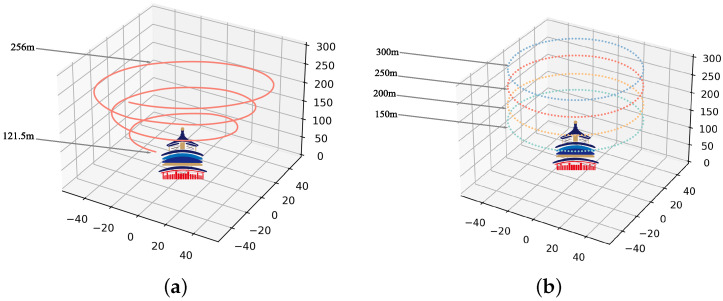
The fight path of the drone. (**a**) University-1652. (**b**) SUES-200.

**Figure 2 sensors-23-00720-f002:**
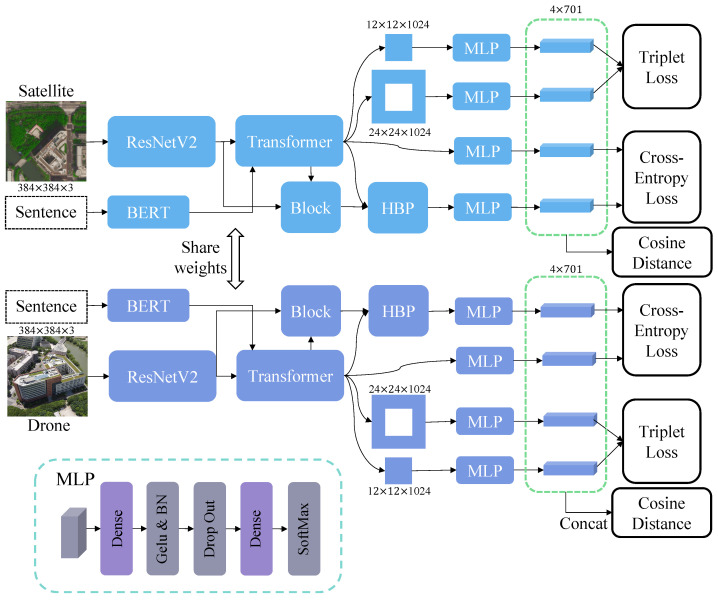
Overview of the proposed method. MBF contains two branches, i.e., the satellite-view branch, and the drone-view branch, respectively, to process different inputs. The above two branches share weights since their patterns are similar. Given one image, ResNet first extracts its feature map. Then, the semantic feature from BERT is input to the Transformer block with the map. (See Figure 3). Next, HBP (See Figure 4) and Partition Strategy are applied to the output of the backbone (Block stands Transformer block). Finally, all these feature descriptors are fed into MLP to calculate loss or distance.

**Figure 3 sensors-23-00720-f003:**
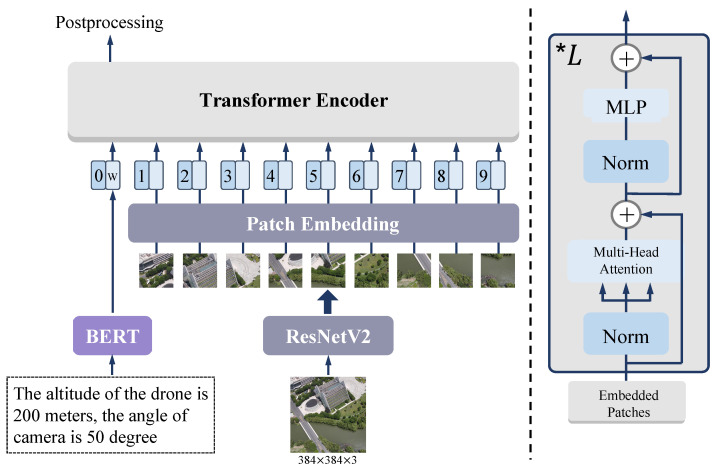
The architecture details of Multimodal block. On the left side of the picture, the descriptive sentence is transformed to word embedding using BERT and aligned with image patch embeddings. Then, those embeddings are fed into Transformer Encoder to extract deeper features. On the right side of the picture, the Transformer block’s detailed architecture is shown.

**Figure 4 sensors-23-00720-f004:**
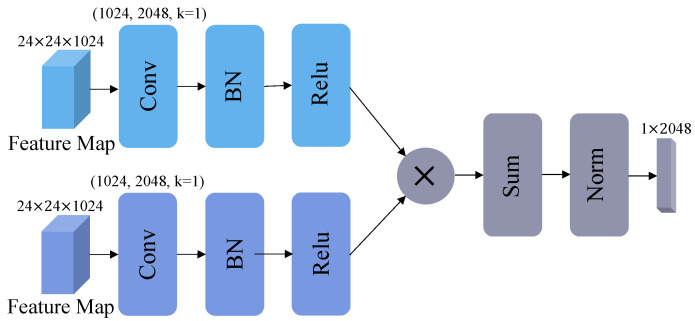
The architecture details of HBP block.

**Figure 5 sensors-23-00720-f005:**
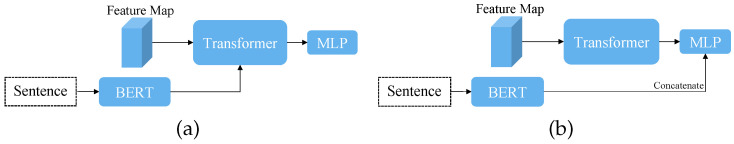
The fusion timing of modals. (**a**) Progressive Fusion. (**b**) Late Fusion.

**Figure 6 sensors-23-00720-f006:**
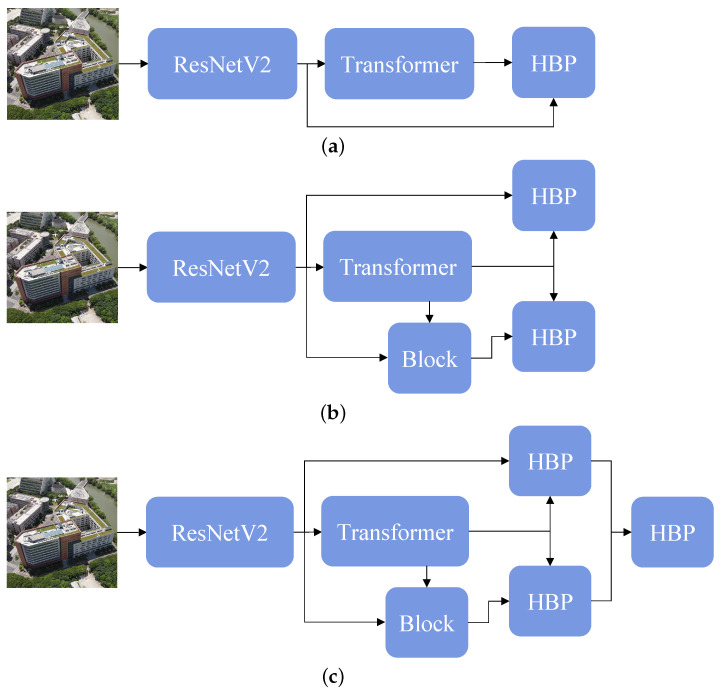
The different designs of HBP block. (**a**) Basic HBP. (**b**) Double HBP. (**c**) Triplet HBP.

**Figure 7 sensors-23-00720-f007:**

Flip padding. We pad 0, 10, 20, 30, and 40 pixels on the left of query images, respectively.

**Figure 8 sensors-23-00720-f008:**
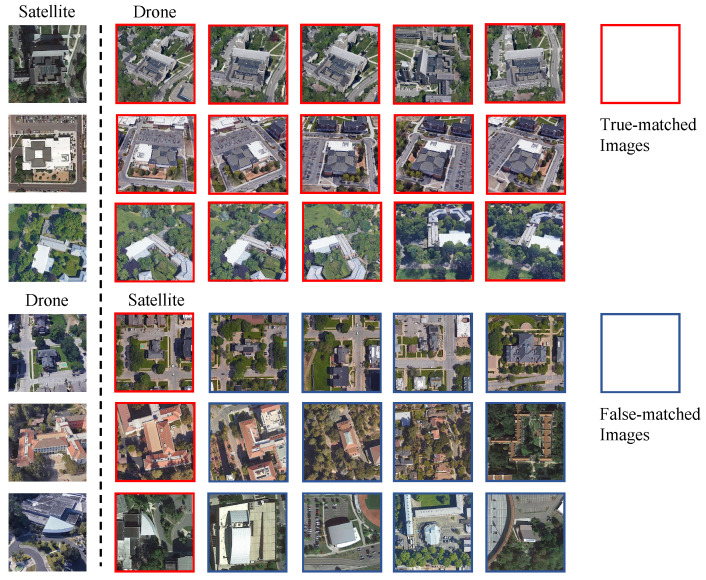
Qualitative image retrieval results.

**Figure 9 sensors-23-00720-f009:**
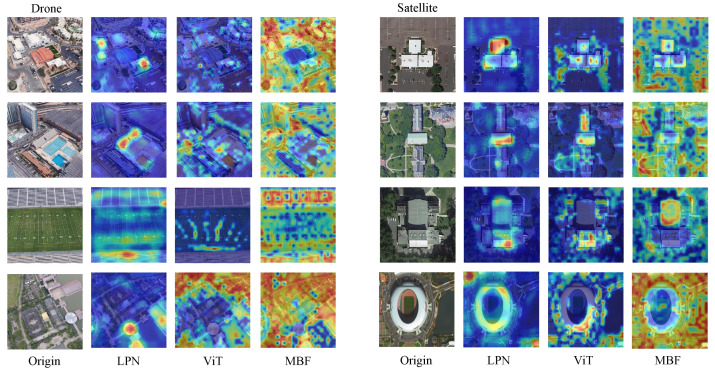
Visualization of heatmaps.

**Table 1 sensors-23-00720-t001:** Overview of geo-localization approaches.

Approach	Author
** Street↔Aerial **	-
CVUSA	Workman et al. [14]
CVACT	Liu et al. [15]
L2LTR	Yang et al. [24]
VIGOR	Zhu et al. [25]
** Drone↔Satellite **	-
University-1652	Zheng et al. [16].
LCM	Ding et al. [17]
LPN	Wang et al. [19]
PCL	Tian et al. [21]
FSRA	Dai et al. [22]
MSBA	Zhuang et al. [20]

**Table 2 sensors-23-00720-t002:** Statistics of University-1652 training and testing sets, including the image number of query set and gallery set.

Training Dataset		
**Views**	**Classes**	**Images**
Drone	701	37,854
Satellite	701	701
**Testing Dataset**		
**Views**	**Classes**	**Images**
Drone Query	701	37,854
Satellite Query	701	701
Drone Gallery	951	51,355
Satellite Gallery	951	951

**Table 3 sensors-23-00720-t003:** Statistics of SUES-200 training and testing sets, including the image number of query set and gallery set.

Training Dataset			
**Views**	**Classes**	**Images at Each Height**	**Images**
Drone	120	6000	24,000
Satellite	120	_	120
**Testing Dataset**			
**Views**	**Classes**	**Images at Each Height**	**Images**
Drone query	80	4000	16,000
Satellite query	80	_	80
Drone gallery	200	10,000	40,000
Satellite gallery	200	_	200

**Table 4 sensors-23-00720-t004:** Comparison with the state-of-the-art in University-1652.

Method	Image Size	Drone→Satellite		Satellite→Drone	
		Recall@1	AP	Recall@1	AP
Baseline [16]	384 × 384	62.99	67.69	75.75	62.09
LCM [17]	384 × 384	66.65	70.82	79.89	65.38
LPN [19]	384 × 384	78.02	80.99	86.16	76.56
LDRVSD [48]	384 × 384	81.02	83.51	89.87	79.80
SGM [49]	256 × 256	82.14	84.72	88.16	81.81
PCL [21]	512 × 512	83.27	87.32	91.78	82.18
FSRA [22]	384 × 384	85.50	87.53	89.73	84.94
MSBA [20]	384 × 384	86.61	88.55	92.15	84.45
MBF	384 × 384	**89.05**	**90.61**	**93.15**	**88.17**

**Table 5 sensors-23-00720-t005:** Comparison with the state-of-the-art in SUES-200.

Drone→Satellite								
**Method**	**150 m**		**200 m**		**250 m**		**300 m**	
	**Recall@1**	**AP**	**Recall@1**	**AP**	**Recall@1**	**AP**	**Recall@1**	**AP**
Baseline [43]	55.65	61.92	66.78	71.55	72.00	76.43	74.05	78.26
ViT [43]	59.32	64.94	62.30	67.22	71.35	75.48	77.17	80.67
LCM [17]	43.42	49.65	49.42	55.91	54.47	60.31	60.43	65.78
LPN [19]	61.58	67.23	70.85	75.96	80.38	83.80	81.47	84.53
MBF	**85.62**	**88.21**	**87.43**	**90.02**	**90.65**	**92.53**	**92.12**	**93.63**
Satellite→Drone								
**Method**	**150 m**		**200 m**		**250 m**		**300 m**	
	**Recall@1**	**AP**	**Recall@1**	**AP**	**Recall@1**	**AP**	**Recall@1**	**AP**
Baseline [43]	75.00	55.46	85.00	66.05	86.25	69.94	88.75	74.46
ViT [43]	82.50	58.88	87.50	62.48	90.00	69.91	96.25	84.10
LCM [17]	57.50	38.11	68.75	49.19	72.50	47.94	75.00	59.36
LPN [19]	83.75	66.78	88.75	75.01	92.50	81.34	92.50	85.72
MBF	**88.75**	**84.74**	**91.25**	**89.95**	**93.75**	**90.65**	**96.25**	**91.60**

**Table 6 sensors-23-00720-t006:** Effect of the fusion timing.

Timing	Drone→Satellite		Satellite→Drone	
	Recall@1	AP	Recall@1	AP
Progressive	**86.27**	**88.25**	**91.73**	**85.92**
Late	85.65	87.71	90.58	85.28

**Table 7 sensors-23-00720-t007:** Effect of the HBP design.

HBP Design	Drone→Satellite		Satellite→Drone	
	Recall@1	AP	Recall@1	AP
Basic HBP	84.55	86.70	90.30	83.97
Double HBP	86.39	88.38	91.58	85.34
Triplet HBP	**86.95**	**88.93**	**91.87**	**86.24**

**Table 8 sensors-23-00720-t008:** Effect of the part feature numbers.

Numbers	Drone→Satellite		Satellite→Drone	
	Recall@1	AP	Recall@1	AP
1	85.96	88.02	91.30	85.93
2	**87.49**	**89.30**	**91.87**	**87.53**
3	87.27	89.13	91.44	87.32
4	87.28	89.16	91.44	87.39

**Table 9 sensors-23-00720-t009:** Results of ablation experiments of the proposed method.

Method	Drone→Satellite		Satellite→Drone	
	Recall@1	AP	Recall@1	AP
Baseline	83.52	85.94	90.30	84.38
+ Triplet Loss	84.63	86.79	90.30	84.39
+ Multimodal	86.27	88.25	91.73	85.92
+ HBP	87.34	89.15	92.87	87.50
+ Part Feature	**89.05**	**90.61**	**93.15**	**88.17**

**Table 10 sensors-23-00720-t010:** Effect of the shifted query.

Shifted Pixel	Drone→Satellite		Satellite→Drone	
	Recall@1	AP	Recall@1	AP
0	89.05	90.61	93.15	88.17
10	88.83	90.44	92.58	88.01
20	87.91	89.69	92.30	87.25
30	86.45	88.50	92.01	86.05
40	84.81	87.11	91.58	84.41

**Table 11 sensors-23-00720-t011:** Matching accuracy of multiple queries.

Query	Drone→Satellite			
	Recall@1	Recall@5	Recall@10	AP
54	93.15	96.58	97.86	94.02
27	93.08	96.65	97.79	93.96
18	92.72	96.81	97.86	93.70
9	92.70	96.81	97.79	93.68
3	91.85	96.67	97.65	92.99
2	91.21	96.42	97.47	92.42
1	89.05	95.76	97.04	90.61

**Table 12 sensors-23-00720-t012:** Comparison with multiple queries in University-1652.

Method	Drone→Satellite			
	Recall@1	Recall@5	Recall@10	AP
Baseline	69.33	86.73	91.16	73.14
LCM [17]	77.89	91.30	94.58	81.05
PCL [21]	91.63	95.46	97.33	90.84
MBF	**93.15**	**96.58**	**97.86**	**94.02**

**Table 13 sensors-23-00720-t013:** Inference time of MBF and other models.

Method	Step 1	Step 2
MBF	2.56s	$3.61\times 10^{-4}$ s
ViT-Base	1.68s	$2.05\times 10^{-4}$ s
LPN [19]	1.01s	$3.24\times 10^{-4}$ s

## Data Availability

Not applicable.

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
