# Peer review of "UAV’s Status Is Worth Considering: A Fusion Representations Matching Method for Geo-Localization"

_sensors, 2023, doi:10.3390/s23020720_

Round 1

Reviewer 1 Report

Few comments as below:

Line 25-- space after 2. similar type of mistakes in entire manuscript.

many abbreviations are missing in entire manuscript

In tables and manuscript -- mention proposed method instead of "ours"

Author Response

\1. Line 25-- space after 2. similar type of mistakes in entire manuscript.

We were really sorry for our careless mistakes. Thank you for your reminder. We also submit the paper for English Editing to polish this manuscript. The English Editing Certificate has been uploaded to the system, too. And we hope the revised manuscript could be acceptable to you.

\2. many abbreviations are missing in entire manuscript

Thank you for your suggestion. We have carefully checked the manuscript and corrected the errors accordingly. And we have added "Abbreviations" in Pg17/18 Ln 451-454 to display all abbreviations in the manuscript.

\3. In tables and manuscript -- mention proposed method instead of "ours"

We have corrected it and we also feel great thanks for your point out. In Pg 10/18 Table 4, Pg 11/18, Table 5, Pg 14/18 Table 12.

Reviewer 2 Report

1. Where does the “Sentence” come from?

2. What is the purpose of semantic information being converted into words embedded in images?

3. Uniform Literature Format

Author Response

\1. Where does the “Sentence” come from?

First, we obtain drone status information such as drone flight height, drone camera angle, etc. by mining the known data in the dataset. For example, the SUES-200 dataset contains information about the drone flight height, drone camera angle. Then use a sentence to summarize this information, for example, “The attitude of the drone is 200 meters, the angle of camera is 50 degree”. Finally, the sentence is sent to BERT to convert to word embedding. In addition, the new adding data in current datasets and the code of experiments will be open source at https://github.com/Reza-Zhu/MBF after the paper is accepted.

We also have added the extra information in SUES-200 in Pg 5/18, Ln 181-183.

\2. What is the purpose of semantic information being converted into words embedded in images?

MBF's backbone network is ViT, so the word embeddings extracted by BERT (the same architecture based on Transformer) can get the same dimension as image embedding, which is easier to concatenate together.

\3. Uniform Literature Format

We feel sorry for our carelessness. After carefully reviewing the journal's reference format. In our resubmitted manuscript, the format is revised.

Reviewer 3 Report

The manuscript entitled "UAV’s Status is Worth Considering: A Fusion Representations
Matching Method for Geo-localization
" has been investigated in detail. The topic addressed in the manuscript is potentially interesting and the manuscript contains some practical meanings, however, there are some issues which should be addressed by the authors:

ü  Highlights should be added to the manuscript.

ü  The authors should work more on the motivation part of the paper and explain their research purpose more properly.

ü  The paper needs to be checked carefully for grammatical mistakes.

ü  Did the authors introduce the details of the dataset?

ü  The authors should discuss future work plans.

ü  The presentation of this paper needs some improvement. For example, some parts have unnecessary empty spaces. The authors should remove these unnecessary empty spaces in the revision.

ü  Figure 8 should be improved.

ü  Could the authors report the running time of the proposed algorithm? In this way, we can justify whether this algorithm can be applied to large-scale dataset.

ü  The authors should address the following limitations. The first limitation is that, the related works should be grouped into two or three subsections. In the current version, the authors all merged them together.

ü  The authors should carefully proofread this paper and correct all the typos in the revision. In the current version, there are still some typos/grammar errors.

Author Response

Comments from reviewer:

The manuscript entitled "UAV’s Status is Worth Considering: A Fusion Representations Matching Method for Geo-localization" has been investigated in detail. The topic addressed in the manuscript is potentially interesting and the manuscript contains some practical meanings, however, there are some issues which should be addressed by the authors.

Reply: We thank the reviewer for the kind comments.

\1. Highlights should be added to the manuscript.

Reply: Thank you for your suggestion. In order to highlight the manuscript, we have added the suggested content to the manuscript in the abstract(Pg 1/18, Ln 5-19) to clarify our contributions. We also have shown the state-of-the-art results at the end of the abstract and introduction. Furthermore, the contributions of this manuscript in Pg 4/18 Ln 141-155, have been updated more clearly.

\2. The authors should work more on the motivation part of the paper and explain their research purpose more properly.

Reply: We have discussed the limitations of existing methods and extended the motivations of our method in Pg 2/18 Ln 43-64, Introduction.

\3. The paper needs to be checked carefully for grammatical mistakes.

Reply: We tried our best to improve the manuscript and made some changes to the manuscript. We also submit the paper for English Editing to polish this manuscript. The English Editing Certificate has been uploaded to the system, too. And we hope the revised manuscript could be acceptable to you.

\4. Did the authors introduce the details of the dataset?

Reply: Yes, we did. In Pg 4/18, Section 2 Dataset, we have detailed the datasets which are applied in our experiments, University-1652 and SUES-200. In order to distinguish the differences between the above datasets, we have added two figures to clarify them in Pg 5/18, Figure 1.

\5. The authors should discuss future work plans.

Reply: Thank you for the suggestions. We have added the future work plans in Pg 16/18, Ln 417-422, Section 5 Discussion. "Future work will focus on collecting a comprehensive multimodal geo-localization dataset for the community. This dataset contains more information about drone status, satellite status, and even weather conditions. Then, we could apply multimodal-based model on it to further improve the performance of model in various environments. On the other hand, the inference time of current model is another big concern. We also will work on reducing the parameters of backbone to faster the inference speed. "

\6. The presentation of this paper needs some improvement. For example, some parts have unnecessary empty spaces. The authors should remove these unnecessary empty spaces in the revision.

Reply: We were really sorry for our careless mistakes. Thank you for your reminder.

\7. Figure 8 should be improved.

Reply: We have revised Figure 8(Current Figure 9 in Pg 16/18). We have added the original ViT-base model to compare with MBF. We also have added images from SUES-200 dataset to improve the diversity of the heat map(Please check the fourth line of Figure 9).

\8. Could the authors report the running time of the proposed algorithm? In this way, we can justify whether this algorithm can be applied to large-scale dataset.

Reply: Thank you for your advice. In our updated manuscript, we have added the experiment in Pg 4/18, Section 4.6, Inference Time, to report the running time of MBF. We also have reported other two models to compare the inference speed. We have noticed the inference time is a concern for MBF. Therefore, the future work plan about reducing the parameters of the model have been mentioned in Pg 16/18, Ln 420-422, Discussion.

\9. The authors should address the following limitations. The first limitation is that, the related works should be grouped into two or three subsections. In the current version, the authors all merged them together.

Reply: Thank you for pointing this out. We have divided the Related Work into 3 subsections: Geo-localization Method, Multimodal, and Pooling in Pg 1-4/18.

\10. The authors should carefully proofread this paper and correct all the typos in the revision. In the current version, there are still some typos/grammar errors.

Reply: We have carefully checked the manuscript and corrected the errors accordingly.